# Knowledge mobilisation: an exploratory qualitative interview study to confirm and envision modification of lay and practitioner eczema mindlines to improve consultation experiences and self-management in primary care in the UK

Fiona Cowdell

Faculty of Health Education and Life Sciences, Birmingham City University, Birmingham, UK

**Correspondence to**
Professor Fiona Cowdell; fiona.cowdell@bcu.ac.uk

## ABSTRACT

**Objectives** To investigate whether initial eczema mindlines, 'collectively reinforced, internalised, tacit guidelines', are an accurate representation of the experiences of lay people and practitioners in primary care and to explore how these mindlines may best be revised to improve eczema care.

**Design** Exploratory qualitative interviews with constant comparative analysis and data mining.

**Setting** UK, primary care.

**Participants** People with eczema or parents of children with eczema (n=19) and primary care practitioners (n=13).

**Results** Interview data were analysed using constant comparison of new data with existing initial eczema mindlines to identify areas of agreement and disagreement. Data were mined for participant's thoughts about *whose* mindlines should be modified, *how* this may be achieved and *what* core content is essential. Eczema mindlines and the spiral of knowledge creation, from which they evolved, intuitively made sense. Participants offered examples of how their eczema knowledge is continually produced and transformed as they interact with others. They reported diverse and wide-ranging influences on their thinking and recognised the critical relationship between lay and practitioner mindlines. For this reason they advocated modifying lay and practitioner mindlines in parallel. Participants advised amendment based on consistent information directed to all who influence eczema care. Information should come from trusted sources and be easy to access, distilled, practical, contextually relevant and amenable to assimilation.

**Conclusions** The purpose here is to improve primary care consultation experiences and self-management in eczema. The remaining challenge is to find novel, simple and pragmatic methods of modifying eczema mindlines to instil shared and consistent understanding. Given the prevalence of eczema and the scope of people who influence self-care, interventions should transcend patient-practitioner boundaries and address the wider community. One conceptually congruent approach is to create a *Ba*,

### Strengths and limitations of this study

► Novel study confirming and extending existing lay and practitioner eczema mindlines.
► Diverse sample of UK patients and parents of children with eczema and primary care practitioners.
► Enhanced understanding of *whose* mindlines should be modified and *when* and *how* offers a starting point for the complex process of mindline amendment.
► Combining of parent and adult with eczema data and data from different practitioner groups may have limited recognition of nuanced differences in mindlines.

which in this case would be a virtual space for generating and sharing eczema knowledge.

## INTRODUCTION

Eczema is a common and burdensome long-term skin condition[1] predominantly treated in primary care.[2] People seek advice from general practitioners (GPs), nurses, health visitors and pharmacy staff. Consultations with GPs can be frustrating for both patient[3] and practitioner[4 5] and there may be considerable differences between beliefs about assessment and treatment.[6] Nurse consultations tend to be more positively regarded.[7 8] Research into the role of pharmacy staff is limited.[9]

The essence of eczema treatment in primary care is the regular application of emollients, at least daily and often for many years, with or without intermittent use of topical corticosteroids (TCS). Non-adherence is common due to high self-management demand[10] and limited, sometimes conflicting, information.[11]

Effective self-management can improve disease outcomes and quality of life.[12] Interventions to support eczema self-management are poorly understood, are not available to all, can be resource intensive and have varying impact.[13] Eczema is not formally categorised as a long-term condition.[14]

Knowledge mobilisation (KM) is concerned with the creation and movement of new knowledge to where it can be most useful.[15] To be effective KM activity must be relational, constructed from social interaction and context-specific.[16–18] One method of mobilising knowledge is amending mindlines. Mindlines were developed from the work of Polanyi[19] and Nonaka & Takeuchi[20] specifically the concept of the **S**ocialisation, **E**xternalisation, **C**ombination, **I**nternalisation (SECI) spiral in which knowledge is perpetually produced and transformed as users interact, collaborate and learn.[17] Mindlines are 'collectively reinforced, internalised tacit guidelines'. They lie beneath clinical decision-making[21] and are underpinned by acceptance that tacit knowledge (knowledge that is implicit and hard to simply transfer to another person) is a more powerful influencer of action than formal codified knowledge (knowledge that can easily be articulated, accessed and shared through mainstream approaches such as documents, educational videos and the like).[19 20] Mindlines are predicated on the notion of a flexible, embodied and intersubjective understanding of knowledge, the existence of multiple realities and acceptance of knowledge being context-specific.[17] In their seminal work Gabbay and le May identified mindlines among practitioners in primary care[21], this and subsequent work has been frequently cited. However, minimal investigation has been undertaken on condition-specific mindlines or the existence of the lay (patient) equivalent of mindlines.[22]

A recent ethnographic study investigated the construction of healthcare practitioner and lay eczema mindlines in primary care.[3 23] From this, initial models of eczema mindlines were developed. For practitioners eczema was a low priority condition for which the '*recipe doesn't change*'. Their mindlines were built on brief and limited early career education, general and focused internet searches, local formularies and guidance, tacit knowledge and interactions with patients and, to some extent, other practitioners. Differences in mindline development were noted between professions.[23] On the whole they perceived limited need to update their eczema knowledge. The initial model of lay (person with eczema or caring for a child with the condition) mindlines indicated that, like practitioners, their knowledge was built up over time from multiple sources. They comprise interactions with family, friends and the wider community and with practitioners, general and focused internet searches, media coverage, trial and error and tacit knowledge. The desire for knowledge was most marked when people were self-managing by default, those who had become disillusioned and had disengaged from primary care.[3]

The purpose of the study reported here is to: (i) investigate whether the initial practitioner and lay mindlines are an accurate representation of the realities experienced by practitioners and lay people in primary care and (ii) consider how eczema mindlines may best be revised or modified by adding reliable and useful knowledge and by erasing outdated or inaccurate information thus potentially improve quality of eczema consultations and self-management.

## METHODS

### Aims

To: (i) investigate whether initial eczema mindlines, developed from previous research by the author of this paper,[3 23] are an accurate representation of the realities experienced by practitioners and lay people in primary care and (ii) consider how they may best be revised or modified.

### Design

Exploratory qualitative interviews[24] with constant comparative data analysis.[25]

### Setting, participants and process

Data were collected by the author, a nurse and researcher, from December 2017 to May 2018. Practitioners were recruited via the local clinical research network and word of mouth and lay participants through a higher education institute website with mass sharing via word of mouth and media posts in an effort to recruit hard-to-reach groups who may not realise their value in this type of study.[26] Maximum variation purposive sampling was applied[27], characteristics were broad and focused on participant's ability and willingness to discuss eczema mindlines by virtue of their knowledge and experience. Single, individual, semi-structured, audio-taped interviews using a topic guide (box 1) were conducted. The topic guide was based on previous research by the author[3 23] and questions likely to elicit thoughtful and detailed responses to address the research questions. Participants comprised lay people (n=19, table 1) and practitioners with between 1 and 26 years clinical experience (n=13, table 2) and lasted from 16 to 55 min. Interviews continued to the point of data sufficiency, that is until the depth and detail in the data gathered fully addressed the research questions.[28] Most interviews were face-to-face, in the participant's

---

> **Box 1  Interview topic guide**
>
> At the beginning of each interview the concept of mindlines and illustrations were discussed.
> ► How does this eczema mindline makes sense to you?
> ► What are the similarities and differences in the way in which you know and think about eczema?
> ► How could eczema mindlines best be amended?
> ► Do you think there are particular information that needs to be added or erased from existing mindlines?

**Table 1** Demographic details of lay interview participants

| Self/child | Age | Gender |
| --- | --- | --- |
| Self | 28 | Female |
| Self | 36 | Male |
| Child | 6 | Female |
| Self | 53 | Male |
| Self | 34 | Female |
| Children | 13 and 15 | Female |
| Self | 75 | Female |
| Self | 26 | Female |
| Self | 49 | Male |
| Self | 28 | Female |
| Self | 39 | Female |
| Child | 5 | Female |
| Self | 24 | Male |
| Child | 7 | Female |
| Children | 17 and 20 | Female |
| Self | 22 | Female |
| Self | 56 | Male |
| Self | 38 | Male |
| Self | 48 | Female |

workplace, and a small number via telephone. In preparation, participants were sent a brief explanation of mindlines and illustrations of the relevant initial eczema mindline.[3 23]

### Data analysis

Data collection and analysis were iterative processes.[29 30] Audio-data was professionally transcribed and transcripts proofread against recordings for accuracy. Data were analysed in two ways. First, in a constant comparison

**Table 2** Demographic details of practitioner interview participants

| Role | Gender |
| --- | --- |
| Dermatology nurse in primary care | Female |
| Dermatology nurse in primary care | Female |
| Community pharmacist | Female |
| Trainee general practitioner | Male |
| Practice nurse | Female |
| Nurse practitioner | Male |
| General practitioner | Male |
| General practitioner | Male |
| General practitioner | Female |
| General practitioner | Male |
| General practitioner | Male |
| Practice nurse | Female |
| Practice nurse | Female |

approach[25] incoming data was compared with data from the two previous studies from which initial eczema mindlines had been developed.[3 23] Here the focus was on identifying similarities and differences in the ways that mindlines are generated, embedded and transformed over time. Second, through subsequent readings of the transcripts, data were mined for participant's thoughts about how mindlines may best be amended and their beliefs about specific information that should be added or erased.

### Reflexivity

A reflexive attitude was maintained throughout, acknowledging own subjectivity and positioning as a nurse and skin health researcher and the influence that this may have on the study.[31]

### Patient and public involvement

Lay people were involved in developing the research question and designing the study. Results will be communicated to participants in a brief summary.

### RESULTS

The results are presented in four sections. First, the extent to which lay and then practitioner eczema mindlines were perceived as being an accurate representation of their realities is considered. Areas of individual similarity and difference are considered. Second, areas of connection are identified, followed by examples of synergy and dissonance between lay and practitioner mindlines. Finally, a synthesis of *whose* mindlines should be amended with *what* information and *how* this could best be achieved is offered.

### Are lay eczema mindlines a reasonable representation of realities?

Lay participants reported that the concept of mindlines intuitively made sense and noted broad congruence in sources of knowledge and beliefs. They reviewed each element of the existing lay eczema mindline, that is: interactions with family, friends and the wider community and with practitioners, general and focused internet searches, media coverage, trial and error and tacit knowledge within the context of being content to self-manage or accept practitioner management or self-managing by default. Lay participants spoke at length about the presented mindline and provided examples and nuanced insights into the how, why and when their mindlines had been constructed.

Interactions with family and friends and the wider community were experienced by all. The most influential information was advice from mothers. Others reported picking up '*snippets*' of information, sometimes unbidden '*I've had someone on the Tube recommending a moisturiser*' (Female, 22). Suggestions could be overwhelming '*they were telling me their regimes or oh try this cream. Everybody had a cream to recommend and it got to the point where I was*

*using so many things, you didn't know what was working from what wasn't'* (Female, 27) and often unfounded *'hearsay… 'oh well so and so tried this', 'oh I know someone who did this''* (Female, 28). Advice from practitioners was sometimes externally validated with other lay people before making a decision about treatment use.

Participants mainly spoke about GP consultations which were frequently viewed as unsatisfactory. GPs were perceived as having limited expertise *'sometimes I get the impression that they're doing no more than you would by Googling it'* (Female, 34) and working on a trial and error basis *'it's generally a bit of chuck everything at the wall and see what sticks'* (Male, 36). In contrast to the initial mindline the pathway of knowledge between lay people and practitioners was perceived as one-way. People experienced their views and experiences as being undervalued and suggested that they were provided with standard, often unhelpful, information, *'I think doctors think everyone's stupid …… I know the advice is well meant but it's generalistic'* (Male, 56). Some mentioned power differentials as the cause of their expertise being dismissed *'I interact with them as equal but he couldn't have that'* (Male, 53).

Lay participants also consulted with nurses, pharmacy staff, health visitors and a physician's assistant. These meetings were more positively evaluated, for example after seeing a nurse *'I felt much more confident that they know in detail what they were dealing with …… they seemed to have a lot more knowledge about products'* (Female, 27). Pharmacists were another source of practical and personalised advice *'I saw the pharmacist at the practice she was actually very, very helpful, she came out with this keep the creams in the fridge, stroke them on ……… the two saviours were the pharmacists really'* (Female, 75).

Most lay participants had searched the internet, the focus was invariably on finding new treatments or cure rather than understanding the condition. Most had used trusted sources such as NHS Choices, but for some this was *'a lovely set of generic blandishments'* (Male, 36). Many had 'Googled' but were selective in using this information, *'lot of self-censoring, you know a filter'* (Female, 34). Finding reliable information electronically was challenging and some withdrew *'I'm wary of Google in case you get frightened by things that you don't need to see'* (Female, 34).

Other media sources, including leaflets, newspapers and magazines, television programmes and social media, were used to a lesser degree and also treated judiciously. Few people read patient information leaflets enclosed with medications describing them as of little practical value. Leaflets in GP surgeries and pharmacies were infrequently taken and poorly evaluated. A few read newspapers and magazines but were sceptical *'I'm quite an avid newsreader, periodically there are articles … don't always believe 100% because I think there could be something sponsored'* (Female, 39). Participants reported seeing occasional television programmes that influenced their thinking but not often their actions; the BBC was considered most trustworthy. Participant's valued social media for example Facebook and Twitter, the value here was perceived reality,

*'a fellow sufferer saying 'this works for me''* [Female, 34]. As with other knowledge sources participants filtered information, *'the critical eye that I have definitely makes me check and double-check, are these valid sources and cross referenced?'* (Female, 26).

Trial and error was a familiar and frustrating experience for every lay participant *'you build up this medicine cabinet full of every ointment and then you have a set of stories about each one'* (Male, 53). Tacit knowledge was viewed as an amalgamation of personal experience accrued over time and hearsay from unknown or unremembered sources often from the wider community. Universal tacit knowledge was that TCS *'thin the skin'* and should be avoided as far as possible. No participants could trace how this thought had entered their mindline, *'I just grew up with that 'oh you have to be careful with this''* (Female, children 13 & 15) and *'umm … I'm trying to think who told me'* (Female 34).

Participants identified with being content to self-manage or accept practitioner management or self-managing by default. Most described moving back and forth between these states, for example, at times of eczema flare, treatment failure or unsatisfactory consultations. These circumstances triggered fresh attempts to find new ways of managing their eczema sometimes with clouded judgement, *'we just jump to unreliable sources or do irrational things because you can be tempted to, when you've got a really bad flare-up'* (Female, 26).

Eczema mindlines instinctively made sense to lay participants and the content presented resonated with their thoughts, beliefs and experiences. They recognised the multiple influences, including the power of social interactions. Their analysis of personal eczema mindlines reinforced the complexity and difficulty in unravelling how all the elements coalesce particularly as, although content is similar for each individual, the way in which mindlines have developed over time had subtle variations. Lay participants linked efforts to amend mindlines closely with different phases of self-management and for this reason suggested that this should be integrated into the mindlines image (see figure 1).

### Are practitioner eczema mindlines a reasonable representation of realities?

Practitioners agreed with the initial mindline but offered less discussion, for example *'so I think it's nice model actually, I think it's all fair, everything you've said is fair and reasonable and appropriate'* (Male GP). All agreed that eczema is not perceived as a priority in primary care, that treatment is limited by their local emollient formulary and that they exercised caution in prescribing TCS. A dominant influence on the mindlines of the four practitioners with self-reported expertise in eczema care was exposure to dermatology experts in secondary care. The trusted and structured guidance given had become firmly embedded leading to increased knowledge, skills and confidence and *'braver'* prescribing of non-formulary emollients (when clinically appropriate) and TCS. These experts contested the backdrop of low priority, but concurred

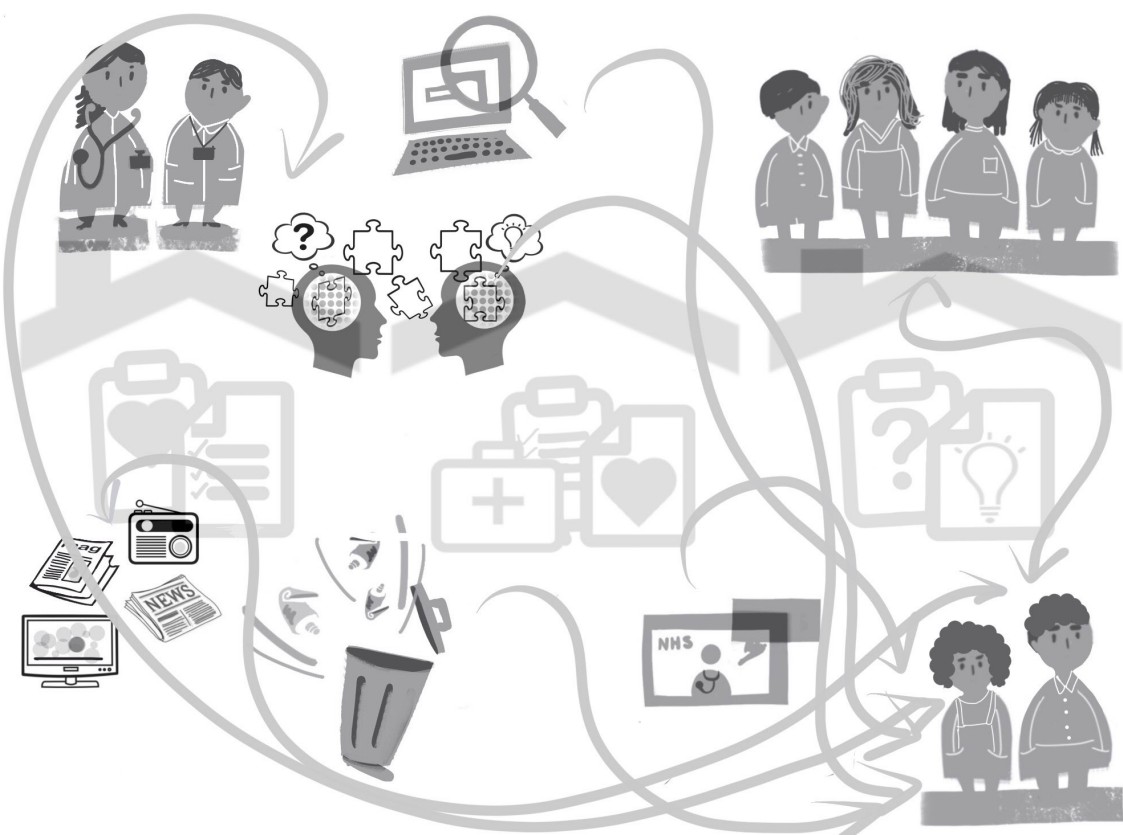

**Figure 1** Revised lay eczema mindline.

with the relative straightforwardness of treatment regimens while viewing these not as a 'recipe' but as principles that should be individualised. The practitioner mindline illustration was amended for clarity (figure 2).

### Areas of lay and practitioner mindline connection

Unprompted participants considered mindlines of the 'other' and the interplay between these entities. Lay participants suggested a need for change in practitioner mindlines but focused on amending lay mindlines. Practitioners were certain about the best targets for mindline amendment. GPs stated that their own and lay mindlines should be amended but considered the former unlikely due to the low priority given to eczema. This view was reinforced by an expert nurse, '*GPs are entrenched in their views*' and a lay participant, '*it's a challenge with the entrenched stubbornness of the medical profession, people will not change their beliefs once they 'know something*'' (Lay male, 36). Other practitioners advised that health visitors, nurses and community pharmacy staff should be targeted given their relative accessibility for patients.

A challenge for all participants was the volume of often contradictory information from a multitude of sources available to lay people. A GP recounted trying to persuade a patient to use TCS properly, '*I validate it with something and say, look I do extra work in dermatology, please just trust me, this is fine*'. However his efforts were thwarted by the

patient's existing mindlines and standard advice given by pharmacists which generated fear.

Practitioners almost universally identified the key reliable and useful knowledge that should be added to, and outdated or inaccurate information that should be erased from, existing mindlines. Core content is summarised in quotations from participants (box 2). Lay participants were less certain about the essential content but agreed key issues included understanding and accepting that eczema may be a long-term condition requiring ongoing treatment with emollients and that TCS have a place in the treatment of flares.

### Examples of synergy and dissonance between lay and practitioner mindlines

Returning to the original definition of mindlines as 'collectively reinforced, internalised tacit guidelines' participants demonstrated both the complexity in development and the influential interplay between lay and practitioner mindlines. Participants identified areas of synergy and dissonance across and between mindlines. A key finding is that eczema knowledge does not involve just people living with the condition and practitioners but rather there is a pervasive social element, for example the received wisdom or '*old wives tale*' that TCSs '*thin the skin*' causing irreparable damage.

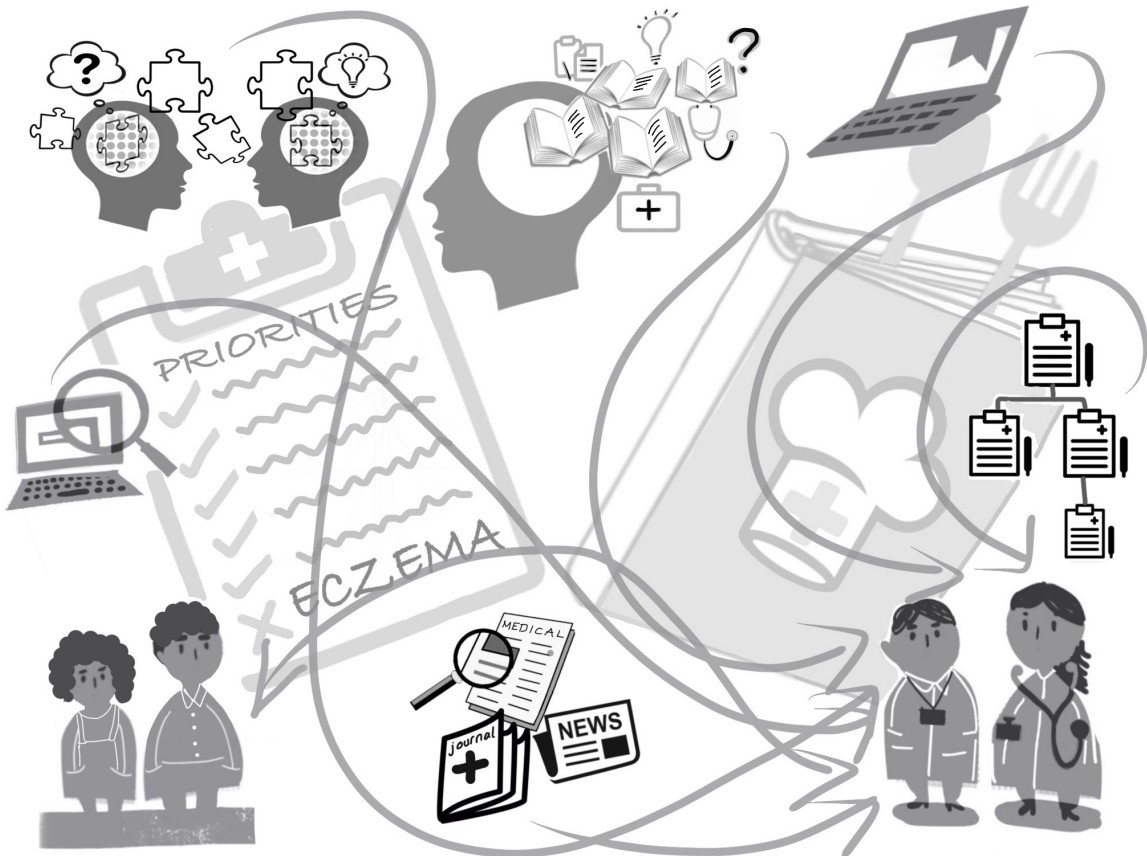

**Figure 2** Revised practitioner eczema mindline.

Perceived need to update was low among most practitioners and this pervaded some lay mindlines '*there's no point in wasting the doctors time*' (Male, 53). As a result eczema was sometimes relegated to an additional concern rather than the primary reason for consultation and then necessarily given limited attention by practitioners, thus reinforcing its low priority. Practitioners reported '*we don't often think about it as a chronic condition which needs regular follow-up and a defined plan*' (Male, trainee GP), leading to reactive treatment with few follow-up appointments. Lack of maintenance therapy impacted on self-management leading to a cycle of flare management rather than a long-term, agreed management plan.

### Mindline amendment: whose, what and how?

Mindline amendment proved a conundrum to all. Lay participants reflected on the futility of simply providing

> **Box 2  Core eczema mindline content from practitioners**
>
> ► '*Never let anyone say to you again that eczema is just a bit of dry skin*'.
> ► '*Treat as a long-term condition*'.
> ► '*Eczema needs maintenance treatment*'.
> ► '*Mantra of the best emollient is the one that (is used)*'.
> ► '*Don't be frightened (of) steroids and use them properly*'.
> ► '*When it does flare-up you've got to hit it hard quickly with your steroid*'.

more information. Immediate suggestions including predictable approaches such as websites, media and leaflets were made but rapidly dismissed. For example, '*website … but who would go on it?*' (Male, 53) and '*leaflets in GP surgeries, but then you just get information fatigue*' (Male, 49). Similarly suggestions about using social media and 'apps' were voiced and rejected within the same sentence. Interactive fora were vetoed due to the need for leadership and moderation. One participant favoured a '*Facebook group specifically for eczema care, I'd probably be part of that… good to have a safe place for people to talk*' (Female, 22) although she acknowledged that sustained participation was unlikely.

Pivotal factors in lay mindline amendment included 'realness', exemplified by '*that's the stuff that I'm interested in and that I'll home in on are the forums where you've got people telling their stories*' (Male, 53) and '*people work off stories don't they, like, I'm living proof*' (Male, 36). Other people's stories were persuasive for some and disliked by others. Trustworthiness, ease of access and applicability to self were important, '*quick trusted and easy to access*' (Female, 34) from trusted sources '*pharmacies, a credible source of information*' (Female, children 17 & 20). All lay participants disliked the generic nature of current information '*eczema is very particular, people regard it as part of who they are, because it does shape what you're doing*' (Male, 53). Disillusionment and lack of motivation to change was identified as a challenge to mindline amendment. Interactions and

relationships were addressed on several levels. Engagement was one hurdle '*you need a way of hooking people in, people who are likely to walk past and go 'I haven't got time to care about that'*' (Female, 34). The impact of social and relational factors suggested a need for wider amendment '*an effective way would be to normalise this thing, there's still a shame element around it, bringing it into normal conversations, everyday discussions, so it's not a thing that you're just hiding away*' (Female, 26). Shared, consistent information could be powerful in eradicating influential '*old wives tales*'.

Practitioners proposed existing educational mechanisms for amending mindlines including: free taught sessions with '*really, really practical, emollient testing*' (Female, nurse) and information which can readily be applied in everyday practice, free online education, sessions during protected learning time. Practitioners rarely accessed available eczema education and were '*wary of messages from pharma or perceived messages from pharma*' (Female, GP). Expert practitioners were exasperated by the lack of motivation for change among generalist colleagues and the continued provision '*outdated and wrong*' advice (Female, nurse). Practitioners were aware of existing patient-focused resources, for example NHS Choices, patient.co.uk, the Primary Care Dermatology Society and the British Association of Dermatologists Patient Information Leaflets, but rarely signposted or used these in consultations.

Participants pointed to the need for shared understandings and consistent advice through synergistic mindlines. Reflection on mindlines brought to the fore the diversity and magnitude of influencing factors and the extent to which knowledge was accessed in different circumstances, for example during an eczema flare. The need for consistency of information was a high priority and one best addressed by seeking to modify the multiple channels of information that contribute to lay and practitioner eczema mindlines in primary care with straightforward, evidence based messages.

## DISCUSSION

The first aim of this study was to investigate whether initial eczema mindlines are an accurate representation of the experiences of practitioners and lay people in primary care. Mindlines intuitively made sense to, and were confirmed by, all participants. Areas of synergy and dissonance were identified, as was the interplay between lay and practitioner eczema mindlines. Second, mindline amendment was investigated. The need to modify and unify mindlines by adding reliable and useful knowledge and by erasing outdated or inaccurate information was universally agreed. However, strategies to achieve amendment proved elusive. The prevalence of eczema, the diversity of *how* and *when* mindlines are developed suggests a need to find focused strategies for wide-reaching, societal amendment to ensure simple, consistent messages to improve consultation experiences and self-management in primary care. Practitioners offered suggestions about essential areas for amendment.

This study is one of few to apply mindline theory to a specific condition and is original in identifying the relationship between lay and practitioner mindlines. The study conforms to the conventions of qualitative research in being rigorous (externally auditable thorough clear reporting), relevant (enhancing understanding of the subject), resonant with reader's experiences and understandings and reflexive.[31] The study has deepened understanding of *whose* mindlines should be modified and *when* and *how* this may best be achieved. This offers a robust starting point for the complex amendment process. Limitations of the study include parent and adult with eczema data being combined, potentially limiting nuanced differences in mindlines. Similarly practitioner mindlines have been viewed as a whole. That said, mindlines are individual and can change over time.

To date little attention has been given to the construction of lay (or patient) mindlines. Gabbay and Le May[17] suggest a patient equivalent, but this possibility is poorly represented in other literature.[22] The term clientlines appears in one study[32] but is not fully explored. However, the notion that knowledge creation inevitably happens in patients is not contested[33] and is extended here in the breadth of influences on lay eczema mindlines. This could be attributable to prevalence, with most people knowing someone with eczema and the perception that it is a 'health problem which is not an illness'[34] and therefore perceived as open to treatment suggestions from anyone.

Amendment of mindlines is rarely reported, a synthesis of 10 years of literature (n=340) identified 28 related to practitioner amendment. These emphasise the importance of collaborative learning, relationship building and effective leadership in the development of valid, collective, evidence-based mindlines.[22] Repeating the search strategy used for this review in August 2018 revealed an abundance of new literature (n=422) but still little directly addressing how mindlines may best be amended. Allied literature indicates the need for a judicious approach to amending practitioner mindlines. Primary care practitioners are necessarily expert generalists[35] and have to know about many conditions. It is clearly impossible for any individual to process all available information. Practitioners use coping strategies such as 'satisficing' that is, curtailing the amount of information gathered to enable them to make a 'good enough' decision.[36] Given the view that eczema is straightforward with treatment being standard and unchanging over the years, motivation to change practice is limited.[23] Likewise treatment decisions are apparently made rapidly, perhaps fitting somewhere between fast, automatic, System 1 thinking and the more deliberative, conscious slow and effortful System 2 approach[37] and instead accessing knowledge from a well-worn path. The danger here is that *this* path may not lead to the best outcomes.

Mindlines evolved from the work of Polanyi and Nonaka & Takeuchi, in particular, the SECI spiral[17] in which knowledge is perpetually produced and

transformed as users interact, collaborate and learn.[38] Gabbay and le May[21] point to the critical nature of knowledge-in-practice-in-context, in which in each context new knowledge is converted by complex social processes. This current study concurs, finding that eczema knowledge is founded in a wider community,[39 40] sometimes facilitated by others[41] and often influenced by numerous, interacting personal and social factors.[42] To date moves to improve eczema self-management and consultation experiences largely comprise educational and psychological interventions.[13 43] This study offers an alternative socially-mediated approach. Returning to the work of Nonaka offers new insights into mindline amendment drawing on the relationship between the SECI spiral and *Ba*. *Ba*, originally discussed by the Japanese philosopher Nishida[44 45] and later refined by Shimizu,[46] is a shared space for knowledge generation and sharing. *Ba* is not about the space per se,[47] rather it is a shared context-in-motion which advances over time and has no fixed boundaries or membership. It is about the here-and-now and thrives on people with different viewpoints.[48] *Ba* space occurs naturally, is not prescribed, relationships are fluid, permeable and iterative.[49 50] It has four characteristics: originating (face-to-face, individual communications), interacting (face-to-face group dialogue), cyber (virtual group interactions) and exercising (virtual individual interactions).[50] In the main *Ba* has been considered in business organisations.[51–56] More recently attention has been given to its application in other contexts, for example in making 'big data' widely useful[57] and in supporting educational endeavours.[50 58 59] To date there is no evidence of attempting to create *Ba* across lay-practitioner-community boundaries. Given the need to amend mindlines, both lay and practitioner, across the community the notion of deliberatively creating *Ba* and integrating innovative approaches to addressing each of the four characteristics in a specific geographical area offers a new opportunity for eczema knowledge mobilisation. This work is currently underway and will be the focus of future publications.

In some respects core treatment of eczema in primary care is relatively straightforward. Experts in the field propose 'getting control' and 'keeping control'[60] which requires emollient maintenance to reduce incidence and severity of flares and need for sufficient use of the right potency TCS when required. The key to effective self-management is supporting people to do these two things well and addressing the enormous challenges they face on a day-to-day enduring basis.[10 61–63]

This study offers a new approach to improving consultation experiences and self-management through active and community-wide mindline amendment to develop consistent and shared understandings. Core reliable and useful knowledge that needs to be integrated, and outdated and inaccurate knowledge to be erased is identified. Mechanisms for change need to be socially-mediated, multi-faceted, tailored to specific groups and convey consistent, evidence based, core messages.

**Acknowledgements** Thanks to Jay Nolan-Latchford for the mindline illustrations and to all the participants who shared their time and thoughts.

**Contributors** FC is the sole contributor to this paper.

**Funding** This article presents independent research funded by the National Institute for Health Research (NIHR). The views expressed are those of the author and not necessarily those of the NHS, the NIHR or the Department of Health.

**Competing interests** None declared.

**Patient consent for publication** Not required.

**Ethics approval** The study was approved by a University Ethics Committee. Written consent was taken for interviews and participants consented to publication of anonymised quotes.

**Provenance and peer review** Not commissioned; externally peer reviewed.

**Data sharing statement** The data sets generated and/or analysed during the current study are not publicly available as they are not designed to be re-analysed by others but are available from the corresponding author on reasonable request.

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
