## [Reviewer comments · BMJ Open]

ARTICLE DETAILS

TITLE (PROVISIONAL)	Knowledge mobilisation: an exploratory qualitative interview study to confirm and envision modification of lay and practitioner eczema mindlines to improve consultation experiences and self-management in primary care in the United Kingdom
AUTHORS	Cowdell, Fiona

VERSION 1 - REVIEW

REVIEWER	Pauline Nelson Alliance Manchester Business School, University of Manchester, UK
REVIEW RETURNED	07-Dec-2018

GENERAL COMMENTS	This is a very interesting paper and an novel application of mindline theory to eczema. It's a well-conducted well-presented study. I had a few comments/questions: * In the Introduction, third paragraph, it would be useful to explain that mindlines work has traditionally been done in clinician/professional study populations and not patient ones - it's implicit but could be made more obvious to help readers follow the argument/rationale for the study.* In the Introduction, at the beginning of paragraph 4 the recent ethnographic study needs referencing at the end of the first sentence (ref 23). Also are there some details missing from the actual reference e.g. author initial, year of submission?* I would be interested to know how the topic guide was developed.* The rationale for the sampling strategy is unclear to me - for example, on what characteristics was the purposive sampling approach based and why did the author include parents of children as well as adults affected with eczema? They are very different groups.* I felt there needed to be a few more details given on the data analytic approach to really understand how the author analysed the data.* In terms of Results - I found these a bit hard to follow as presented. Perhaps numbering the sections i) and ii) as mentioned at the very beginning of the Results might help. One of the strengths of the paper is the lay/professional comparison of data but it seemed as though some sections had lay/professional analysis together and others were separated. I think it needs to be a comparison all the way through and the similarities and
---

	differences between lay/professional perspectives drawn out a bit more to help the reader grasp the main issues. * I didn't get a sense of any similarities and differences in the data between affected adults and the parents of affected children. There would be differences I imagine as these are distinct groups and it would be important to address this. * I also missed any distinction between perspectives of the different professional groups included - they all have different training and approaches to clinical management so I would imagine had different view of the mindline. * Could the author include a section on study limitations - if it is there I missed it.
--	---

REVIEWER	Dr Emma Teasdale University of Southampton, UK
REVIEW RETURNED	26-Mar-2019

GENERAL COMMENTS	This is a thoughtful, well-conducted qualitative study exploring lay and practitioner mindlines in relation to eczema. On the whole it is a clear and interesting manuscript. I would, however, recommend the following amendments: Page 2 line 43 – ‘to’ is missing from the sentence ‘The purpose of this work is’ Page 4 line 42 – It would be useful to define or provide examples of what you mean by tacit and codified knowledge to improve the readability of the manuscript. Page 4 line 55 - Please state in the text whether the ‘recent ethnographic study’ was previous work carried out by the author or not as it is not quite clear. It would also be useful to clarify this in the aims i.e. investigate whether initial eczema mindlines developed by the author thorough previous research Page 6 line 3 – please can you clarify what is meant by data sufficiency and how this differs from data saturation? Page 8 line 59 – should it read ‘These circumstances triggered’? Page 14 line 19 – you mention ‘creating Ba and integrating innovative approaches to addressing each of the four characteristics in a specific geographic area offers a new opportunity for eczema knowledge mobilisation’. Could you clarify this further by explaining or providing examples of how this might be achieved?
--

VERSION 1 – AUTHOR RESPONSE

Reviewer: 1 Reviewer Name: Pauline Nelson	
This is a very interesting paper and a novel application of mindline theory to eczema. It's a well-conducted well-presented study. I had a few comments/questions:	Thank you, I really appreciate your positive feedback

* In the Introduction, third paragraph, it would be useful to explain that mindlines work has traditionally been done in clinician/professional study populations and not patient ones - it's implicit but could be made more obvious to help readers follow the argument/rationale for the study.	I've made this point more clearly in para 3 However, minimal investigation has been undertaken on condition specific mindlines or the existence of the lay (patient) equivalent of mindlines (22).
* In the Introduction, at the beginning of paragraph 4 the recent ethnographic study needs referencing at the end of the first sentence (ref 23). Also are there some details missing from the actual reference e.g. author initial, year of submission?	I've added reference I'm just waiting for the editors decision on this manuscript – hopefully it will be accepted very shortly
* I would be interested to know how the topic guide was developed.	I have added The topic guide was based on previous research by the author (3, 23) and questions likely to elicit thoughtful and detailed responses to address the research questions.
* The rationale for the sampling strategy is unclear to me - for example, on what characteristics was the purposive sampling approach based and why did the author include parents of children as well as adults affected with eczema? They are very different groups.	I have added Maximum variation purposive sampling was applied (27); characteristics were broad and focused on participant's ability and willingness to discuss eczema mindlines by virtue of their knowledge and experience. I agree they are very different groups and have added this as a limitation
* I felt there needed to be a few more details given on the data analytic approach to really understand how the author analysed the data.	Thank you, good point. I have added Data analysis Data collection and analysis were iterative processes (29, 30). Audio-data was professionally transcribed and transcripts proof read against recordings for accuracy. Data were analysed in two ways. Firstly in a constant comparison approach (25) incoming data was compared with data from the two previous studies from which initial eczema mindlines had been developed (3, 23). Here the focus was on identifying similarities and differences in the ways that mindlines are generated, embedded and transformed over time. Secondly, through subsequent readings of the transcripts, data were mined for participant's thoughts about how mindlines may best be amended and their beliefs about specific information that should be added or erased.
* In terms of Results - I found these a bit hard to follow as presented. Perhaps numbering the sections i) and ii) as mentioned at the very beginning of the Results might help. One of the strengths of the paper is the lay/professional comparison of data but it seemed as though some sections had lay/professional analysis together and others were separated. I think it needs to be a comparison all the way through	Thank you, I take your point and have signposted better and changed headings in results. The results are presented in four sections. Firstly the extent to which lay and then practitioner eczema mindlines were perceived as being an accurate representation of their realities is considered. Areas of individual similarity and difference are considered.

and the similarities and differences between lay/professional perspectives drawn out a bit more to help the reader grasp the main issues.	Secondly areas of connection are identified, followed by examples of synergy and dissonance between lay and practitioner mindlines. Finally a synthesis of whose mindlines should be amended with what information and how this could best be achieved is offered. I am reluctant to change the order of the results section as it represents the complexity well.
* I didn't get a sense of any similarities and differences in the data between affected adults and the parents of affected children. There would be differences I imagine as these are distinct groups and it would be important to address this.	An interesting point. I have added this as a limitation although similarities and differences tended not to be dichotomised in this way.
* I also missed any distinction between perspectives of the different professional groups included - they all have different training and approaches to clinical management so I would imagine had different view of the mindline.	Unexpectedly all participants had similar views in this study – but differences have come out in some of my other work. I haven't mentioned this again as I think it just adds too much here and I hope that people who want to know more detail may refer to previous and future papers.
* Could the author include a section on study limitations - if it is there I missed it.	Added as indicated above
Reviewer: 2 Reviewer Name: Dr Emma Teasdale	
This is a thoughtful, well-conducted qualitative study exploring lay and practitioner mindlines in relation to eczema. On the whole it is a clear and interesting manuscript. I would, however, recommend the following amendments:	Thank you, again I appreciate this positive evaluation of my work
Page 2 line 43 – 'to' is missing from the sentence 'The purpose of this work is'	Thank you, corrected
Page 4 line 42 – It would be useful to define or provide examples of what you mean by tacit and codified knowledge to improve the readability of the manuscript.	I've added brief definitions for clarity, Mindlines are "collectively reinforced, internalised tacit guidelines". They lie beneath clinical decision-making (21) and are underpinned by acceptance that tacit knowledge (knowledge that is implicit and hard to simply transfer to another person) is a more powerful influencer of action than formal codified knowledge (knowledge that can easily be articulated, accessed and shared through mainstream approaches such as documents, educational videos and the like) (19, 20).
Page 4 line 55 - Please state in the text whether the 'recent ethnographic study' was previous work carried out by the author or not as it is not quite clear. It would also be useful to clarify this in the aims i.e. investigate whether initial eczema mindlines developed by the author thorough previous research	Thanks, I've just amended in the aims as it looks a bit clunky if I change 4, 55 Aims To: (i) investigate whether initial eczema mindlines, developed from previous research by the author of this paper (3, 23), are an accurate representation of the realities experienced by practitioners and lay people in primary care and

	(ii) consider how they may best be revised or modified.
Page 6 line 3 – please can you clarify what is meant by data sufficiency and how this differs from data saturation?	I've added clarification. Interviews continued to the point of data sufficiency; that is until the depth and detail in the data gathered fully addressed the research questions (28). I'm never convinced by saturation, you could go on getting new ideas, data forever – so I've used this pragmatic alternative. O'Reilly M, Parker N. 'Unsatisfactory Saturation': a critical exploration of the notion of saturated sample sizes in qualitative research. Qual Res. 2013; 13:190-197.
Page 8 line 59 – should it read 'These circumstances triggered'?	Thank you, corrected
Page 14 line 19 – you mention 'creating Ba and integrating innovative approaches to addressing each of the four characteristics in a specific geographic area offers a new opportunity for eczema knowledge mobilisation'. Could you clarify this further by explaining or providing examples of how this might be achieved?	Good point, if I explained in detail it would be too long for this paper. I've added "This work is currently underway and will be the focus of future publications".

VERSION 2 – REVIEW

REVIEWER	Pauline Nelson Alliance Manchester Business School, University of Manchester, UK
REVIEW RETURNED	26-Apr-2019

GENERAL COMMENTS	The author has addressed all my points satisfactorily - thank you. It's a very nice paper!
--

REVIEWER	Dr Emma Teasdale University of Southampton, UK
REVIEW RETURNED	15-Apr-2019

GENERAL COMMENTS	All my previous comments have been addressed - many thanks. I have no further comments to add.
--